# Vascular Risk Factors in Ischemic Stroke Survivors: A Retrospective Study in Catalonia, Spain

**DOI:** 10.3390/healthcare10112219

**Published:** 2022-11-05

**Authors:** Silvia Reverté-Villarroya, Rosa Suñer-Soler, Jose Zaragoza-Brunet, Gisela Martín-Ozaeta, Patricia Esteve-Belloch, Iago Payo-Froiz, Esther Sauras-Colón, Fidel Lopez-Espuela

**Affiliations:** 1Nursing Department, Universitat Rovira Virgili, Avenue Remolins, 13-15, 43500 Tortosa, Spain; 2Hospital de Tortosa Verge de la Cinta, ICS, IISPV, C/Esplanetes no 14, 43500 Tortosa, Spain; 3Faculty of Nursing, University of Girona, Emili Grahit, 77, 17071 Girona, Spain; 4Metabolic Bone Diseases Research Group, Nursing Department, Occupational Therapy College, University of Extremadura, Avenue of the University s/n, 10003 Cáceres, Spain

**Keywords:** ischemic stroke, survivors, epidemiology, risk factors, sex differences, rural population

## Abstract

Background: The distribution of vascular risk factors (VRFs) and stroke management vary by geographic area. Our aim was to examine the percentage of the VRFs according to age and sex in ischemic stroke survivors in a geographical area on the Mediterranean coast of Southern Catalonia, Spain. Methods: This was a multicenter, observational, retrospective, community-based study of a cohort, the data of which we obtained from digital clinical records of the Catalan Institute of Health. The study included all patients with a confirmed diagnosis of ischemic stroke who were treated between 1 January 2011 and 31 December 2020. Patients met the following inclusion criteria: residing in the study area, age ≥ 18 years, and presenting ≥1 modifiable vascular risk factor. The exclusion criteria were as follows: death patients (non-survivors) and patients without modifiable VRFs. We collected the demographic, clinical, and VRF variables of the total of 2054 cases included, and we analyzed the data according to age groups, sex, and number of VRFs. Results: Most of the patients included were in the 55–80 age group (*n* = 1139; 55.45%). Of the patients, 56.48% (*n* = 1160) presented ≤ 2 modifiable VRFs, and the age group <55 years old (67.01%) presented more VRFs. Hypertension and (>80 years old (38.82%)) and dyslipidemia (<55 years (28.33%)) were the most prevalent VRFs. In the age group 55–80 (69.59% men), the prevalence of VRFs was higher ((3–4 VRF (42.76%) and >4 VRF (5.35%)). Conclusions: These results suggest the presence of many VRFs in people diagnosed with ischemic stroke—although with a lower percentage compared to other studies—and the need for specific individualized interventions for the control of modifiable RFs related to primary and secondary prevention of stroke.

## 1. Introduction

In Spain, patients with ischemic stroke have a higher risk of cardiovascular recurrences [1,2], and it has been shown that 90.5% of these strokes are attributable to modifiable vascular risk factors (VRFs), which affect their prognosis and, consequently, highlight the importance of detecting and managing these VRFs [3]. Moreover, differences among individuals with diagnosed vascular disease are considerable, resulting from combined characteristics, such as age, genetics, VRFs, the effectiveness of preventive measures, treatment, and life expectancy [4].

The prevalence of the different VRFs and the health strategies for stroke management vary worldwide; hence, we must not only be careful when extrapolating data from other countries or geographical areas, but we must also learn about the epidemiological situation and specific characteristics of each region when assessing preventive and treatment strategies [5]. Catalonia is an autonomous region of 7,758,615 inhabitants [6], with around 14,018 stroke discharges/year, of which 45.3% are women [7]. However, the influence of age, sex, and factors associated with ischemic stroke is still inconsistent in the existing literature, and data on temporal trends in differences by sex and age are scarce, which encourages further research in this line. The objective of this study was to examine the distribution of the VRFs according to age and sex in ischemic stroke survivors in a geographical area on the Mediterranean coast of Southern Catalonia, Spain.

## 2. Materials and Methods

This is an observational, multicenter, retrospective, community-based study conducted on patients diagnosed with ischemic stroke in a rural environment (Terres de l’Ebre Health Region, Catalonia, Spain) (Figure 1), which includes 11 primary healthcare centers and a hospital (Hospital de Tortosa Verge de la Cinta (HTVC)), with second level care, which is the reference center for managing patients with acute stroke and tending to the neurological care needs of a population of 180,383 inhabitants [6,7].

The study included all patients with a confirmed diagnosis (data relating to the stroke disease including classical vascular risk factors, neuroimaging characteristics (CT scan and/or MRI), and the final etiopathogenic diagnosis of stroke) of ischemic stroke who were treated at the HTVC between 1 January 2011 and 31 December 2020. Patients met the following inclusion criteria: residing in the study area, age ≥ 18 years, and presenting ≥1 modifiable vascular risk factor. The exclusion criteria were as follows: death patients (non-survivors) and patients without modifiable VRFs. We selected cases based on medical records coded with the 10th version of the International Classification of Diseases (ICD-10) [8] from the e-SAP database of the Catalan Institute of Health (ICS). The Information Department of the Terres de l’Ebre Territorial Management of the ICS performed an automated extraction of the minimum basic data set (MBDS) of hospital discharges and the Integrated Electronic Prescription System (SIRE).

We collected diagnosis of ischemic stroke (codes I63, I63.1, I63.2, I63.3, I63.4, I63.5, I63.6, I63.7, I63.8, I63.9) and the presence of modifiable vascular risk factors: hypertension (HPN) (codes I10, I15.9), hyperlipidemia (codes E78.2, E78.4, E78.49, E78.5), heart disease (codes I21, I21.9, I25, I25.2), atrial fibrillation (AF) (codes I48.0, I48.1, I48.2, I48.3, I48.4, I48. 91, I48.92, I49.01), hyperglycemia and diabetes mellitus (DM) (codes R73.9, E11), overweight and obesity (codes E66, E66.2, E66.3, E66.8, E66. 9), and toxic habits (active alcohol consumption and smoking) (codes F10.10, F10.19, F10.20, F10.23, F10.28, F10.92 and Z72.0, F17, F17.2, F17.21). We registered demographic characteristics, namely age, sex, and residence, and we also compared vascular risk factors and demographic data between 2011–2015 and 2016–2020. Two 5-year cutoffs were performed for the analysis due to changes in the diagnosis and management of stroke, mainly because in 2015 approval was obtained for the use of mechanical thrombectomy in Spain, a fact that modified the care of patients with ischemic stroke [9].

The data were included in an ad hoc repository, which was delivered to the principal investigator in a completely anonymous, supervised format compliant with Organic Law 3/2018 on personal data protection. As this was a retrospective and big data study, it did not require patient consent, and only required approval of the person legally responsible for the transfer of the HTVC, which was duly obtained.

The Strengthening the Reporting of Observational Studies in Epidemiology (STROBE) guidelines were followed in this observational study.

### Data Analysis

A descriptive analysis was performed using absolute values to indicate the admission rate, frequency, and percentage for the qualitative variables and the median and interquartile range for the quantitative variables. The analysis was carried out by stratifying the sample by age groups (<55, 55–80, and >80 years). Likewise, as it is a 10-year-long cohort, the demographic and diagnostic (Dx) analysis of the population was divided into two temporary groups of 5 years each (diagnosed in 2011–2015 and 2016–2020, both included). We evaluated the differences between both groups using the chi-square and Mann–Whitney U tests for qualitative and quantitative variables, respectively. All analyses were performed using SPSS 27.0 statistical software (IBM, Universidad Rovira Virgili license), and values of *p* < 0.05 were considered significant.

## 3. Results

The present study included 2054 cases registered as ischemic stroke survivors, 56.33% of which were men. The median age of diagnosis of stroke in these patients was 76, with an interquartile range of 67–83. The age group with the most patients was 55–80 years old, consisting of 1139 (55.45%) of the total sample. Out of the total population studied, 1160 (56.48%) presented at least two modifiable vascular risk factors, mostly HPN (35.86%), dyslipidemia (26.07%), heart disease (15.10%), and overweight and obesity (12.58%), followed by hyperglycemia and DM (6.28%), smoking (2.63%), and alcoholism (1.48%) (Table 1). Upon comparing the diagnosis groups, no significant differences were observed, except in the age of diagnosis 75 (65–82) against 77 (68–84) (*p* < 0.001), with age being higher in the group diagnosed between 2016 and 2020 (Table 1).

We collected and analyzed data on a total of 4986 VRFs per age group (<55, 55–80, and >80 years). In all age groups, the two VRFs with the highest percentage were hypertension, with the highest among the age group of >80 years (38.82%), and dyslipidemia, with the highest percentage in the group of <55 years (28.33%). However, in the group of <55 years, the third most prevalent VRF was overweight and obesity (17.68%), unlike other age groups where the third VRF was heart disease (13.89% in the age group 55–80 and 19.17% in >80 years), ahead of overweight and obesity (13.54% in 55–80 years and 9.62% in >80 years). Likewise, the percentage of alcoholism was higher in the age group of <55 years (3.39%) compared to the rest of the groups (1.79% in 55–80 years and 0.48% in >80 years) (Table 2).

We grouped the number of VRFs present into three groups (1–2, 3–4, and >4) and subsequently related each to age groups. We observed that most patients presented between one and two VRFs, with the age group of <55 years old showing the highest percentage (67.01%). Furthermore, the age group between 55 and 80 years old showed the highest prevalence of VRFs (42.76% presented between 3 and 4, and 5.35% presented >4 VRFs) (Table 3).

Regarding the distribution of men and women in each of the age groups, it is worth noting that in the age groups <55 years and 55–80, the percentage of men is higher (69.59% and 64.63%, respectively), while this percentage is reversed in the age group >80 years, which comprises more women (60.33%). However, the distribution of the number of VRFs was quite similar in both men and women, with a higher percentage of the group 1–2 vascular risk factors in all age groups, and the age group 55–80 years showing the highest prevalence of VRFs (6.39% men and 3.47% women presented >4 VRFs) (Table 4).

## 4. Discussions

For this study, we examined the percentage of the VRFs according to age and sex in patients with ischemic stroke in the Terres de l’Ebre healthcare region, in the autonomous region of Catalonia, Spain.

Out of the 2054 cases studied, more than half presented at least two modifiable VRFs, with a greater presence of HPN, dyslipidemia, heart disease, and overweight and obesity. Moreover, we observed that the age group 55–80 showed the highest incidence of VRFs. Upon comparing these data against the data extracted from the last Stroke Audit, carried out by the Master Plan for Cerebral Vascular Disease of Catalonia (PDMVC) [10], we observed a mean age four years higher in this study than the overall mean of the last Audit, and a similar percentage of individuals over 80 years of age [11]. Additionally, we noticed significant differences in the age of diagnosis, which was higher between the years 2016 and 2020 in individuals over 80 years of age compared to the previous period. This finding coincides chronologically with the approval of neuro-interventional treatment [9], which broadened the therapeutic window of acute care for survivors of ischemic stroke and extended the age of treatment, which could have partly influenced the increase in the mean age of patients treated in the last five years [12], together with quality healthcare and better management of VRFs by the population with a history of stroke [13].

Four out of ten patients analyzed were women, data similar to those published in this latest PDMVC audit [10]. However, the proportion of women is still lower, in line with other studies [14]. Moreover, it has been observed that as women age, their risk equals the risk of men of the same age, which is reflected in the decrease in the ratios of the male–female incidence rates [15]. On the one hand, epidemiological studies have shown that women have a lower risk of stroke than men when studied in younger adult populations; however, on the other hand, the risk for women is significantly higher after reaching 80 years old, which could be explained by the higher survival rate of women [14,16]. Nonetheless, other authors have indicated that women have a higher risk of stroke throughout their lives and more recurrences, partly due to the longer life expectancy of women and because age is not a modifiable VRF [17]. Thus, women live to ages at which the risk of stroke is higher, but after having a stroke, the mortality rate is lower and they experience more favorable hospital and functional recovery [18].

Regarding VRFs, hypertension is the most frequent modifiable RF, but in contrast to the data from the PDMVC and other studies, this VRF was observed in a lower proportion [19]. In this study, a third of the patients had hypertension, while practically twice as many participants in other studies had this diagnosis [20,21,22]. These differences, including differences observed in other VRFs, could be explained by the different registration systems used and that we only included patients from a specific geographical area of Catalonia, who could have lower rates of VRFs. Specifically, from the second half of the 20th century onwards, the mortality rate from vascular diseases in people in this area in the province of Tarragona decreased, along with its associated risk factors, due to measures related to health promotion and improvement of health conditions, security, etc. [23]. In individuals aged 60 years or older, higher proportions of hypertensive persons with acute stroke have also been observed in the same geographical area, as well as higher percentages of the other risk factors assessed [24].

When analyzing the differences according to age group, the highest percentage of HPN was observed in the group >80 years old (almost 4 out of 10 participants), and the highest percentage of dyslipidemia was observed in the group <55 years old (almost 3 out of 10 participants). However, in the group <55 years old, the third most prevalent VRF was overweight and obesity (almost 2 out of 10). Previously reported data in primary care in Catalonia showed percentages of approximately 50% for HPN and almost 60% for hypercholesterolemia; the data refer to individuals whose risk factors had also been entered in the electronic medical record, as in this study [25]. In other data reported from the same autonomous region in the Barcelona area, in people aged 55 years or younger, hypertension showed a prevalence of 33%, while smoking ranked first, unlike the present study. In patients over 55 years of age, the most frequent VRF was hypertension (almost 60%), followed by diabetes (23%) [20].

The analysis of the results by sex showed that the number of VRFs was similar in both men and women, with a higher percentage of 1–2 VRFs in all age groups, especially in women under 80 years old. Women in the age group 55–80 had the highest number of RFs (3–4). The linear regression model showed that age groups 55–80 and >80 years old had more VRFs, but no differences were found by sex. Women have unique VRFs for stroke such as pregnancy and hormone therapy. Other VRFs include diabetes, atrial fibrillation, and higher rates of hypertension after age 65 which are likely to increase with the aging of the female population [26,27,28,29].

This study helps expand knowledge regarding ischemic stroke survivors from a specific geographic area, whose men and women have unique characteristics affecting the prevalence of VRFs.

Regarding the limitations of this study, since we analyzed data from patients from a specific non-urban territory in our region, comparing these results with similar data from the rest of the Catalan territory is challenging. Nevertheless, we may consider that this study advances the knowledge of VRFs in specific geographical areas, which could guide more accurate, individualized, and local interventions to prevent stroke, such as the promotion of healthy lifestyle habits, resulting in better management of hypertension, diabetes mellitus, heart disease, dyslipidemia, smoking, and/or obesity among other factors, considering their strong association with the increased risk of stroke mortality [21]. On the other hand, the target population of the present study was ischemic stroke survivors with vascular risk factors, so no data are shown on mortality, survival, or the percentage of patients without vascular risk factors. Likewise, as this was a retrospective study, no causality data can be provided.

## 5. Conclusions

The patient profile is characterized by a mean age of 76 years, more than half of patients having 1–2 VRFs, a higher proportion of people with hypertension and dyslipidemia in all age groups, a higher proportion of heart disease in people aged 55 or older, and overweight and obesity especially in people under 55 years of age. The distribution of the number of VRFs was similar between men and women, although individuals over 55 years old presented more vascular risk factors.

These results suggest the presence of many VRFs in people diagnosed with IS—although with a lower percentage compared to other studies—and the need for specific individualized interventions for the control of modifiable VRFs related to primary and secondary prevention of stroke.

## Figures and Tables

**Figure 1 healthcare-10-02219-f001:**
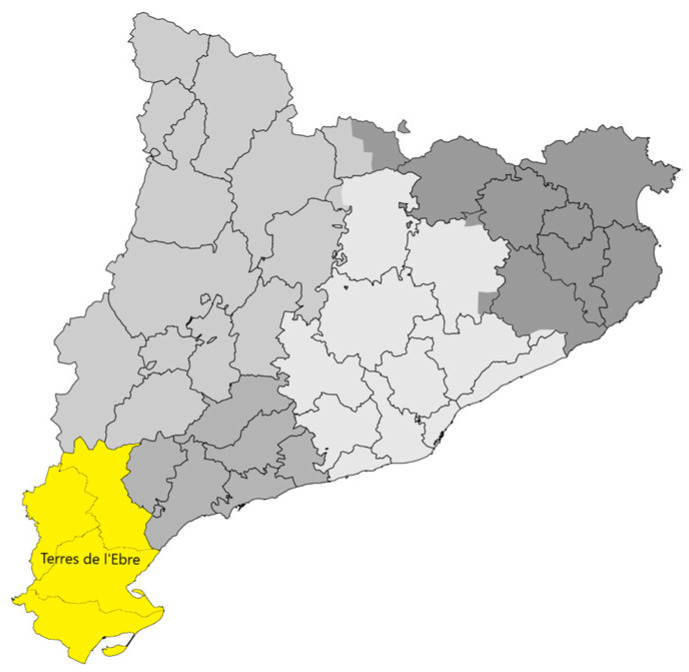
Map of Catalonia, Spain. The Terres de l’Ebre health region.

**Table 1 healthcare-10-02219-t001:** Distribution of demographic characteristics and vascular risk factors of patients according to disease in both year ranges.

	Population (*n* = 2054)	Dx 2011-15 (*n* = 829)	Dx 2016-20 (*n* = 1225)	*p*-Value
**Diagnostic age, years (IQR)**	76 (67–83)	75 (65–82)	77 (68–84)	<0.001
**Age group *n* (%)**				0.013
<55	194 (9.44)	85 (10.25)	109 (8.90)	
55–80	1139 (55.45)	484 (58.38)	655 (53.47)	
>80	721 (35.10)	260 (31.36)	461 (37.63)	
**Sex *n* (%)**				0.962
Men	1157 (56.33)	468 (56.45)	689 (56.24)	
Women	897 (43.67)	361 (43.55)	536 (43.76)	
**No. VRFs *n* (%)**				0.284
1–2	1160 (56.48)	456 (55.01)	704 (57.47)	
3–4	813 (39.58)	344 (41.50)	469 (38.29)	
>4	81 (3.94)	29 (3.50)	52 (4.24)	
**VRF type *n* (%)**				0.243
HPN	1788 (35.86)	725 (35.42)	1063 (36.17)	
Dyslipidemia	1300 (26.07)	535 (26.14)	765 (26.03)	
Heart disease and AF	753 (15.10)	289 (14.12)	464 (15.79)	
Hyperglycemia and DM	313 (6.28)	134 (6.55)	179 (6.09)	
Overweight and obesity	627 (12.58)	270 (13.19)	357 (12.15)	
Alcoholism	74 (1.48)	29 (1.42)	45 (1.53)	
Smoking	131 (2.63)	65 (3.18)	66 (2.25)	

VRF: vascular risk factor; Dx: diagnosis; AF: atrial fibrillation; DM: diabetes mellitus. Qualitative variables are represented by the number of cases (percentage), and the quantitative variables are represented by median (interquartile range). The Mann–Whitney U and chi-square tests were used to assess the differences in quantitative and qualitative variables, respectively, between the two groups.

**Table 2 healthcare-10-02219-t002:** Distribution of vascular risk factors according to age group.

VRF (*n* = 4986) *n* (%)	<55	55–80	>80	*p*-Value
HPN	144 (34.87)	998 (34.31)	646 (38.82)	<0.001
Dyslipidemia	117 (28.33)	766 (26.33)	417 (25.06)	<0.001
Heart disease and AF	30 (7.26)	404 (13.89)	319 (19.17)	<0.001
Hyperglycemia and DM	28 (6.78)	205 (7.05)	80 (4.81)	<0.001
Overweight and obesity	73 (17.68)	394 (13.54)	160 (9.62)	<0.001
Alcoholism	14 (3.39)	52 (1.79)	8 (0.48)	<0.001
Smoking	7 (1.69)	90 (3.09)	34 (2.04)	<0.001

VRF: vascular risk factor; HPN: hypertension; AF: atrial fibrillation; DM: diabetes mellitus. Percentages are calculated concerning the total for each of the age groups (<55, 55–80, and >80 years old). The chi-square test was used to evaluate the differences between the age groups in individuals with each of the VRFs.

**Table 3 healthcare-10-02219-t003:** Relationship between the number of vascular risk factors and age groups.

VRF Number (*n* = 2054) *n* (%)	<55	55–80	>80	*p*-Value
1–2	130 (67.01)	591 (51.89)	439 (60.89)	<0.001
3–4	60 (30.93)	487 (42.76)	266 (36.89)	
>4	4 (2.06)	61 (5.35)	16 (2.22)	

VRF: vascular risk factor. Percentages are calculated based on the total for each age group (<55, 55–80, and >80 years). The chi-square test was used to evaluate the differences between <55 years old and 55–80, *p* < 0.001; <55 and >80, *p* = 0.180; 55–80 and >80, *p* < 0.001.

**Table 4 healthcare-10-02219-t004:** Relationship of risk factors according to age groups, stratifying the sample by sex; men (*n* = 1157) and women (*n* = 897).

Age Groups	<55		*p*	55–80		*p*	>80		*p*
	Men 135 (69.59)	Women 59 (30.41)		Men 736 (64.62)	Women 403 (35.38)		Men 286 (39.67)	Women 435 (60.33)	
1-2 VRFs (*n* = 644)	89 (65.93)	41 (69.49)	0.891	378 (51.36)	213 (52.85)	0.113	177 (61.89)	262 (60.23)	0.121
3-4 VRFs (*n* = 453)	43 (31.85)	17 (28.81)		311 (42.26)	176 (43.67)		99 (34.62)	167 (38.39)	
>4 VRFs (*n* = 60)	3 (2.22)	1 (1.69)		47 (6.39)	14 (3.47)		10 (3.50)	6 (1.38)	

VRF: risk factor. The percentage of the sex variable is calculated for the total of each of the age groups (<55, 55–80, and >80 years), and the percentage of the number of VRFs is calculated concerning the total of men and women within each group. The chi-square test was used to evaluate the differences.

## Data Availability

The data are available upon reasonable request.

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
