# Peer review of "Vascular Risk Factors in Ischemic Stroke Survivors: A Retrospective Study in Catalonia, Spain"

_healthcare, 2022, doi:10.3390/healthcare10112219_

Round 1

Reviewer 1 Report

Thank you for the opportunity to review this article. The authors declared that the aim of this study is to examine the prevalence of the risk factors according to age and sex in ischemic stroke (IS) in a rural area of south Catalonia. The paper is clearly structured, and each chapter is well written, so it is easy to understand what the authors what to transmit. However, there is a crucial point that is not clear to me, and I hope that the authors will be able to clarify it.

As the authors declared in Materials and Methods (line 65) “Patients who died during the study period were excluding”. I invite the author to explain the reason for this choice, because I think that the exclusion of deceased patients significantly changes the outcomes of the analysis. For example, the percentage of patients with hypertension (HPN) is significantly lower than that reported in other studies, and I consider it a relevant aspect because is recognized the relation between HPN and IS mortality. In addition, I believe that some information regarding deceased patients might be relevant, for the purpose of the study. I think the authors need to explain why they made this decision that they need to add this data, or they have to explicit that their research is focused on a population of "stroke survivors".

Author Response

Dear Reviewer,

Thank you very much for your comprehensive review of our manuscript. We, the authors, are very grateful.

Sincerally,

Reviewer 2 Report

The findings presented in this article are undoubtedly highly relevant, as they provide an advance and new data on the prevalence of cardiovascular disease risk factors and, in particular, ischemic stroke in a certain geographical area of Southern Catalonia, Spain.

The study carried out is well designed, scientifically and technically sound. The data are presented in a well-structured form.

The results of the study are consistent with the purpose, the materials and methods of research used are described in detail, the statistical methods are consistent with the purpose of the study.

The article is written in clear language, the data presented in the tables are correct, reflect the scientific results presented in the text.

The conclusions are consistent, logical and supported by the data presented.

Cited references to publications are relevant and appropriate.

The data presented in the manuscript are of undoubted interest for a professional audience with regard to studying the prevalence of risk factors for cardiovascular diseases and, in particular, ischemic stroke, as well as for developing effective methods of treatment, prevention of healthy lifestyle and reduction of cardiovascular disease risk in the population of different age groups of the territorial area under study.

In my opinion, the title of the article requires correction, since it was not the epidemiology of ischemic stroke proper that was studied, but the prevalence of cardiovascular disease risk factors and IS, in particular.

It is also necessary to provide correct data on the number of people living in Catalonia according to the source [6] - Instituto de Estadística de Cataluña Población en Cataluña y por provincias. Available online: 279 https://www.idescat.cat/pub/?id=aec&n=245&lang=es (accessed on 20 april 2022). 

Author Response

(The authors gave the same response as above.)

Reviewer 3 Report

You can look at the document (pdf) comments

Author Response

Dear Reviewer,

Thank you very much for your comprehensive review of our manuscript. We, the authors, are very grateful.

All suggested changes are detailed by means of change control in the attached manuscript.

Sincerally,

Reviewer 4 Report

The aim of present manuscript was to investigate the prevalence of RF among ischemic stroke patients at the Southern coast of Catalonia. The data was collected obtained from digital clinical records of the Catalan Institute of Healt  and therefore it has potential for significant observations.

However there is a major methodological weakness of the study stated on methods on Page 2 line 65 ”Patients who died during the study period were excluded”. If authors really did not include death patients the clinical relevance of present paper is not significant. At the present manuscript the major observation is that the younger the patients were the more risk factor they presented. It is tempting to speculate that the older the patients with multiple risk factor the more likely they had significant burden of atherosclerosis and furthermore they died during the follow-up more likely either on MACE or other causes of death and were not included in the data and analyses. This also refers to conclusions on abstract. Authors admit that the RF were not as frequent as in earlier reports. Also why patients without modifiable Rf were excluded?

Abstract: Aims, results and conclusion should be revised to be more readable. Also, methods should include the above-mentioned exclusion/inclusion criteria.

Introduction: The flow should be revised. The essential topics are in the text, but the order of issues and the flow of the text is not as it should be in a scientific article.

Methods: Please see major concern. E10 modifiable? Flow chart would be helpful especially because of exclusion criteria.

Results: Table 1. Abbreviations should be written out also in the table legend. The rationale of dividing study period is not clear. Does it really provide new scientific data? If then comparation between each year might be more easy to understand an might demonstrate trend or something ?

Table 2 Minor differences and p<0.0001 for each RF? Abbreviations should be written out also in the table legend?

Table 3 Does the math’s really match? Table 2 presents that RF are evenly distributed across all age groups and on would suspect even that the result would be opposite? Abbreviations should be written out also in the table legend

Table 4. P-value only for sex, which age category? Other p-values? Abbreviations should be written out also in the table legend

Discussion: The present data if non-survivors were exclude and also IS patients without modifiable risk factors excluded does not present all IS patients at the region. Therefore the title and discussion should be revised or authors should include these patients into results.

Author Response

(The authors gave the same response as above.)

Reviewer 5 Report

Thank you for the opportunity to review this interesting article.

Peer-review report 1991923

The paper Ischemic stroke epidemiology in Southern Catalonia, Spain written by authors Silvia Reverté-Villarroya, Rosa Suñer-Soler, Jose Zaragoza-Brunet, Gisela Martín-Ozaeta, Patricia Esteve-Belloch, Iago Payo Froiz, Esther Sauras-Colón and Fidel Lopez-Espuela, presents the prevalence of risk factors  (RFs) according to age and sex in ischemic stroke in a 15 geographical area on the Mediterranean coast of Southern Catalonia, Spain.

Below are a few remarks, the clarification of which, would allow a better understanding of the text and complete this interesting study:

1. In the Introduction: The authors presented some conclusions from international studies [1,2,3,4,5] and statistical studies on the local population (Catalonia, Spain) [6,7], but did not justify the necessity of choosing these statistical analyses on the dependence prevalence of risk factors according to age and sex in ischemic stroke in area on the Mediterranean coast of Southern Catalonia, Spain.

2. For clarification, please state the following:

- In lines 154-155 a multiple linear regression model was mentioned, but the data presented in line 156, I believe, refer to a correlation between the parameters number of RFs and the age groups 55-80 years and >80 years. In order to talk about the linear regression, an equation between the mentioned parameters must be specified. I think that the authors should insist on these correlations also through graphical representations that are relevant and that can suggest the types of regressions that could be obtained.

- in the age group 55-80 years it would be interesting to divide this interval (e.g. 55-60 years, 60-70 years; 70-75 years, 75-80 years) to reveal the age group (subgroup) with the highest incidence of ischemic stroke and associated risk factors.

- even if the authors state in line 87 that they used in this observational study STROBE guidelines, I think it would be necessary to check the document STROBE-Statement-Checklist of items that should be included in reports of case-control studies (https://www.strobe-statement.org/) to see if each item of the mentioned document is included in the architecture of this article.

The discussions are very detailed and coherent.

The final conclusions are coherent and summarize the fact that the results of the study suggest the presence of many  risk factors in people diagnosed with ischemic stroke.

The tables are clears and easy to understand.

The paper uses grammatof risk factors  cally and academically appropriate language, of substance, and is easy to understand.

The article is an interesting study, which can be published after incorporating the reviewers' remarks.

Author Response

(The authors gave the same response as above.)

Round 2

Reviewer 3 Report

No new comments, I have seen that the propossed corrections and all my suggestions have been included. It seems to me the manuscript has been clearly improved.

Reviewer 4 Report

No further comments authors have revised manuscript according to the comments.